# Rli51 Attenuates Transcription of the *Listeria* Pathogenicity Island 1 Gene *mpl* and Functions as a *Trans*-Acting sRNA in Intracellular Bacteria

**DOI:** 10.3390/ijms25179380

**Published:** 2024-08-29

**Authors:** Álvaro Morón, Laura Ortiz-Miravalles, Marcos Peñalver, Francisco García-del Portillo, M. Graciela Pucciarelli, Alvaro Darío Ortega

**Affiliations:** 1Department of Cell Biology, Facultad de Ciencias Biológicas, Universidad Complutense de Madrid, 28040 Madrid, Spain; alvmoron@ucm.es; 2Laboratory of Intracellular Bacterial Pathogens, National Centre for Biotechnology (CNB)-CSIC, 28049 Madrid, Spain; lortiz05@ucm.es (L.O.-M.); marcos.pennalver@uam.es (M.P.); fgportillo@cnb.csic.es (F.G.-d.P.); 3Department of Molecular Biology, Universidad Autónoma de Madrid, Centro de Biologia Molecular Severo Ochoa (CBM) CSIC-UAM, 28049 Madrid, Spain

**Keywords:** *Listeria monocytogenes*, sRNA, *trans*-acting, Rli51, *mpl*, LIPI-1, premature transcription termination, iron, surface protein, Lmo2186

## Abstract

*Listeria* pathogenicity island 1 (LIPI-1) is a genetic region containing a cluster of genes essential for virulence of the bacterial pathogen *Listeria monocytogenes*. Main virulence factors in LIPI-1 include long 5′ untranslated regions (5′UTRs), among which is Rli51, a small RNA (sRNA) in the 5′UTR of the Zn-metalloprotease-coding *mpl*. So far, Rli51 function and molecular mechanisms have remained obscure. Here, we show that Rli51 exhibits a dual mechanism of regulation, functioning as a *cis*- and as a *trans*-acting sRNA. Under nutrient-rich conditions, *rli51-mpl* transcription is prematurely terminated, releasing a short 121-nucleotide-long sRNA. Rli51 is predicted to function as a transcription attenuator that can fold into either a terminator or a thermodynamically more stable antiterminator. We show that the sRNA Rli21/RliI binds to a single-stranded RNA loop in Rli51, which is essential to mediate premature transcription termination, suggesting that sRNA binding could stabilize the terminator fold. During intracellular infection, *rli51* transcription is increased, which generates a higher abundance of the short Rli51 sRNA and allows for transcriptional read-through into *mpl*. Comparative intracellular bacterial transcriptomics in *rli51*-null mutants and the wild-type reference strain EGD-e suggests that Rli51 upregulates iron-scavenging proteins and downregulates virulence factors from LIPI-1. MS2 affinity purification confirmed that Rli51 binds transcripts of the heme-binding protein Lmo2186 and Lmo0937 in vivo. These results prove that Rli51 functions as a *trans*-acting sRNA in intracellular bacteria. Our research shows a growth condition-dependent mechanism of regulation for Rli51, preventing unintended *mpl* transcription in extracellular bacteria and regulating genes important for virulence in intracellular bacteria.

## 1. Introduction

Small non-coding RNAs (sRNAs) are bacterial post-transcriptional regulators that control gene expression through varied molecular mechanisms affecting target mRNA/s availability, stability, and/or translation [1]. sRNAs are classically divided into two main categories: *cis*-acting and *trans*-acting, depending on whether the sRNA is encoded within the same locus as its target mRNA(s) or a different locus, respectively. *Cis*-acting sRNAs are either regulatory segments included in untranslated regions (UTRs) or encoded on the opposite strand of target mRNAs (e.g., antisense RNAs and excludons) and regulate exclusively that particular single gene. By contrast, *trans*-acting sRNAs can regulate multiple targets through imperfect binding to the mRNAs [2]. However, cumulative evidence in the last few years revealed a somehow overlooked mechanistic diversity in sRNA-based regulation, which suggests that this basic classification might be limited [3]. 

sRNA-based regulation has been associated with virulence in most model pathogenic bacteria [4,5]. Hints linking sRNA function with virulence have most often relied on two facts. First, the sRNA locus belongs to a genetic segment carrying virulence-associated genes, which is absent in non-pathogenic strains (i.e., a pathogenicity island or PAI) [6,7], and second, the sRNA is induced during the course of the infection of eukaryotic cells [8,9,10,11]. Subsequently, the definite proof was that null mutants for the sRNA showed reduced virulence in in vitro and/or in vivo models of infection. However, the number of sRNAs whose deletion leads to an unequivocal attenuation phenotype is still exceedingly low compared to the number of sRNAs located in PAIs and/or induced in infection conditions. For instance, comparative genome-wide analyses showed that around one-fifth of all identified sRNAs (29 out of 150) in the foodborne pathogen *Listeria monocytogenes* (*L. monocytogenes*) are specifically expressed in intracellular bacteria infecting macrophages. On the other hand, a similar number of sRNA loci were exclusively encoded in pathogenic *L. monocytogenes* strains and were absent in related non-pathogenic *Listeria* species [9]. However, to our knowledge, a solid functional association of *L. monocytogenes* sRNAs with virulence has been established only for a handful of them [8,9,12]. Thus, at present, the precise regulatory role of the majority of sRNAs induced in infection conditions and/or located in PAIs has not been described yet. 

*L. monocytogenes* is a Gram-positive bacterial pathogen that causes listeriosis, one of the most serious foodborne diseases in humans and animals [13]. The most relevant virulence factors are clustered in the genome making up PAIs, from which the *Listeria* pathogenicity island 1 (LIPI-1) is present in most isolates of pathogenic species *L. monocytogenes* and *L. ivanovii* [13]. LIPI-1 is highly induced in intracellular bacteria and comprises seven well-characterized protein-coding genes that are critical for intracellular survival and bacterial dissemination: *prfA*, *plcA*, *hly*, *mpl*, *actA*, *plcB*, and *orfX* [14]. In addition, there are two annotated sRNAs in LIPI-1. Rli74 and Rli51 are located in the 5′-UTR of the virulence factor genes *actA* and *mpl*, respectively [7,8,9]. A single promoter drives the intracellular expression of actA, which encodes a cell surface protein essential for intracellular motility and the invasion of neighboring cells. This promoter generates an mRNA with a long UTR that includes Rli74. Deletion of *rli74* halted ActA-mediated actin recruitment and cell-to-cell spread, indicating that Rli74 is a *cis*-acting sRNA essential to achieve a sufficient ActA level to drive bacterial motility within cell host cytoplasm [15]. Similarly, the transcription start site of *mpl*, which codes for a metalloprotease that activates other important virulence factors, such as the phospholipase PlcB, and is essential for cell-to-cell spread, is located 150 bps upstream of the start codon, suggesting that Rli51 represents a long 5′UTR of the *mpl* transcript [16,17]. Intriguingly, there are no further molecular or functional analyses investigating the role of Rli51 as a *cis*-encoded sRNA. Thus, currently, it is unknown whether Rli51 regulates the expression of the metalloprotease Mpl and the relevance that such *cis* regulation would have for the virulence of *L. monocytogenes*. 

Evidence retrieved from different transcriptomic analyses of *L. monocytogenes* challenges the view of Rli51 as a 5′-UTR of *mpl* transcript. Specifically, tiling arrays suggested that Rli51 could be a short 121-nucleotide-long RNA induced in blood and the intestine of infected mice [8]. Consistently, Term-seq, a genome-wide approach aimed to identify the 3′ end of expressed transcripts, identified a transcription termination for Rli51 upstream of the *mpl* start codon [18]. In fact, the occurrence of Rli51 as a short transcript has been recently verified by northern blot in *L. monocytogenes* grown in PrfA-inducing conditions [19]. Collectively, these observations suggest that, at least in some conditions, the transcription of Rli51 might not be coupled to *mpl* mRNA. Therefore, it is not yet clear if Rli51 sRNA is embedded within the 5′-UTR of *mpl*, is an independent transcript, or undergoes a condition-specific premature transcriptional termination (PTT). Furthermore, we do not know the molecular mechanisms that regulate this process and the functional role of Rli51.

In this work, we aimed to investigate the function of Rli51, an sRNA encoded in the LIPI-1 of *L. monocytogenes*. Analysis of the transcriptional landscape of the *rli51-mpl* locus under nutrient-rich laboratory conditions and in intracellular bacteria indicated the occurrence of a conditional regulation of Rli51 transcriptional read-through into *mpl*. Then, using translational fusions with *egfp,* we identified the sequence elements and potential *trans*-acting factors relevant to Rli51 transcription termination regulation. Finally, we explored the functional role of Rli51 in infection-relevant conditions through a transcriptomic analysis of *rli51* deletion mutants collected from infected epithelial JEG-3 cells. Our work shows that Rli51 does not participate in *mpl* upregulation during the infection of eukaryotic cells. Instead, we found that the LIPI-1-encoded sRNA Rli51 functions as a *trans*-acting sRNA in intracellular bacteria.

## 2. Results

### 2.1. Rli51 Does Not Play a Relevant Role in the Upregulation of mpl during Infection

The 5′ UTR regions of the LIPI-1 genes *hly* and *actA* are essential for achieving the highest expression in infection-relevant conditions [15,20]. Thus, we first hypothesized a role for Rli51 as a *cis*-acting element that contributes to *mpl* upregulation in infection conditions. To test this hypothesis, we analyzed the impact of *rli51* deletion on *mpl* transcript levels. We generated an *rli51*-null mutant in the *L. monocytogenes* 1/2a EGD-e strain genomic background and monitored *mpl* mRNA abundance by reverse transcription coupled to quantitative PCR (RT-qPCR). The transcription of LIPI-1 is induced in intracellular bacteria. Consistently, *mpl* mRNA abundance in bacteria collected from infected cells was increased by more than 100 folds compared to the extracellular condition (Figure 1A). However, deletion of chromosomal *rli51* did not impair *mpl* upregulation in intracellular bacteria; on the contrary, although not significant, in the Δ*rli51* strain *mpl* expression in intracellular bacteria was higher than in the the wild-type strain (Figure 1A). These results indicate that Rli51 does not play a significant role in maximizing *mpl* expression during infection. 

Induction of *mpl* might result from the activation of a cryptic promoter located downstream of the *rli51* promoter, which, in turn, would drive the transcription of an *mpl* transcript without Rli51. Such a situation would make the infection-specific induction of *mpl* independent of Rli51. In such a scenario, Rli51 might still have a *cis*-acting regulatory activity over *mpl* in other conditions. To ascertain the occurrence of such a yet unidentified promoter element upstream of *mpl,* we analyzed *mpl* transcription using translational fusions in which the *rli51* promoter had been deleted (Figure 1B) [16,17,21]. In constructs devoid of the *rli51* promoter, *mpl*-*egfp* expression showed a 10- and 100-fold decrease when grown in BHI and intracellular bacteria, respectively (Figure 1B). Thus, in the absence of the *rli51* promoter, there was not any induction of *mpl* in intracellular bacteria that could be attributed to an additional cryptic promoter. This result indicates that *mpl* upregulation during infection results exclusively from the activation of the *rli51* promoter. Therefore, both in LIPI-1-inducing and in LIPI-1-non-inducing conditions, Mpl protein synthesis originates from a unique *mpl* mRNA, which includes Rli51 as a 5′UTR. 

In addition to transcriptional activation, cellular *trans*-acting factors could concomitantly act upon Rli51 post-transcriptionally to support *mpl* upregulation in intracellular bacteria. To analyze the influence of Rli51 on *mpl* mRNA abundance independently of these potential infection-specific factors, we sought to induce the expression of translational fusions of *mpl*-*egfp* in bacteria grown in BHI (Figure 1C). To accomplish this aim, we used an inducible *tet* promoter that allowed us to achieve a level of *mpl*-*egfp* mRNA in extracellular conditions comparable to that observed with the *rli51-mpl* promoter in intracellular bacteria (Figure 1B,C). Deletion of *rli51* in the construct did not result in a decreased *mpl*-*egfp* transcript abundance, irrespective of the level of induction; instead, the absence of *rli51* even seemed to have a positive effect on *mpl*-*egfp* expression at higher levels of transcript induction (Figure 1C). These results recapitulate the effect that chromosomal *rli51* deletion had on endogenous *mpl* expression in intracellular bacteria (Figure 1A) and confirm that *mpl* upregulation during infection results solely from the induction of the *rli51-mpl* promoter. 

Thus far, we have analyzed the potential effects of Rli51 over the *mpl* transcript. However, Rli51 might be a binding site for post-transcriptional regulators that control Mpl protein expression without affecting its mRNA abundance. To rule out the participation of such regulators, we analyzed the phenotype of the *rli51* deletion mutant in in vitro infection models (Figure 1D). We reasoned that, similar to the previously demonstrated Rli74-mediated *cis*-acting regulation over ActA [15], disrupting a key regulator of the virulence factor Mpl—an essential protease for the activation of the phospholipase PlcB and cell-to-cell spread [22,23]—would significantly affect the infection process. Therefore, we analyzed the bacterial burden at 6 h after infection, which is enough time for *L. monocytogenes* to escape from the vacuole and spread from cell to cell [23]. *rli51* knockout strains showed no significant differences in the intracellular proliferation index (quotient of the number of intracellular bacteria at 6 h and at 2 h after infection) compared to EGD-e infecting the human syncytiotrophoblast (JEG3) or rat fibroblast (NRK-49F) cell lines (Figure 1D). These results suggest that Mpl protein expression levels are not altered in the absence of Rli51 sRNA.

Altogether, these results demonstrate that Rli51 does not participate in the upregulation of *mpl* in intracellular bacteria. Conversely, *mpl* expression seems to be higher in the Δ*rli51* strain, both in extracellular and intracellular bacteria, which suggests that Rli51 might be a *cis*-acting negative regulatory element of the *mpl* transcript.

### 2.2. Condition-Specific Premature Termination of Rli51-mpl Transcription

The results from previous reports suggest that Rli51 occurred as a short 121-nucleotide-long RNA [8,18,19], which goes against the view of Rli51 as the 5′UTR of the *mpl* transcript. To investigate whether Rli51 transcription is decoupled from *mpl* transcription, we quantified endogenous Rli51- and Rli51-*mpl*-containing transcripts by RT-qPCR (Figure 2A, upper panel). With this approach, we verified that Rli51 occurs both as a short transcript (Rli51 from now on) and as the 5′UTR of *mpl* (Rli51-*mpl*) (Figure 2A). Both RNA molecules—Rli51 and Rli51-*mpl*—are expressed in bacteria grown in BHI and in bacteria collected from infected JEG3 cells; however, the proportion of each species was different (Figure 2A). Rli51-*mpl* represented less than 2% of all Rli51 in bacteria grown in BHI, while this proportion scaled up to 35% in intracellular bacteria (Figure 2A, right axis). Thus, in exponentially growing bacteria, almost all Rli51 is represented by a short transcript, while in intracellular bacteria, there is a transcriptional read-through that allows for Rli51-*mpl* transcription. These results suggest that Rli51-*mpl* transcription may be subjected to condition-specific premature transcription termination (PTT).

We sought to verify if Rli51 prompts Rli51-*mpl* transcript premature termination. We utilized *mpl::egfp* translational fusions with the *rli51* promoter in which the *rli51* locus had been replaced by the 5′UTR of the listeriolysin-encoding *hly* gene. The 5′UTR of *hly* is similar to *rli51* in length (100 nt). In addition, it positively regulates the expression of listeriolysin during intracellular infection while having only a moderate effect when grown in broth [20]. Constructs including the *hly* 5′UTR seemed to have an increased *mpl*-*egfp* transcript abundance in intracellular bacteria (Figure 2B), suggesting that Rli51 could halt transcriptional read-through. *rli51* and *mpl* loci are separated by a 30 nt long sequence that includes the ribosome-binding site. This sequence was, however, irrelevant for PTT since swapping it by the equivalent segment in *hly* had no effect compared to the *rli51-mpl::egfp* control construct (Rli51 * vs. Rli51 in Figure 2B). These results suggest that Rli51 is necessary and sufficient to mediate the PTT of Rli51-*mpl* RNA. 

In intracellular bacteria, we observed an increase in both Rli51 and Rli51-*mpl* transcripts, which entailed an asymmetric increase in the abundance ratio of Rli51-*mpl* relative to total Rli51 (Figure 2A). As PrfA-dependent promoters are induced during intracellular infection, we hypothesized that the activation of the *rli51-mpl* promoter is the sole driving force allowing for the transcriptional read-through of Rli51-*mpl* in this condition. To test this hypothesis, we used an inducible *tet* promoter to control the transcription of *rli51-mpl-egfp* translational fusions. Induction of the *tet* promoter in exponentially growing bacteria in BHI to levels comparable to those achieved with the *rli51* promoter in intracellular bacteria led to a consistent and proportional increase in the transcriptional read-through of Rli51-*mpl* (Figure 2C, right axis). Replacement of *rli51* by the 5′UTR of *hly* relieved Rli51-mediated PTT, allowing for transcriptional read-through and *mpl*-*egfp* transcript production (Figure 2C). The difference in the relative levels of the *mpl* transcript between Rli51 and constructs containing the 5′UTR of *hly* was greater at lower levels of induction (Figure 2C). This result indicates that Rli51-mediated PTT is more efficient at a lower transcriptional activity of the promoter. Therefore, Rli51-mediated transcription termination may be alleviated by increasing the transcriptional activity, which would contribute to impeding *mpl* transcription when bacteria are not under LIPI-1-inducing conditions.

### 2.3. Trans-Acting Regulation of Rli51-Mediated Transcriptional Termination

We next aimed to decipher the molecular mechanism by which Rli51 mediates condition-specific PTT. Conditional PTT represents quite a pervasive post-transcriptional control mechanism in Gram-positive bacteria, where a particular *cis*-acting sRNA located in a 5′UTR can fold into two mutually exclusive conformations: a transcriptional terminator that mediates PTT and an antiterminator that allows for reading through into the downstream gene [18,24]. Computational analysis using a recently developed algorithm (*Pasific*) [25] predicted that Rli51 could indeed fold either as terminator or antiterminator conformations (Figure 3A). The terminator fold included three stem loops, and the last one is the actual terminator. In the antiterminator fold, this terminator stem loop is distorted by being involved in the formation of a 20-nucleotide-long stem together with the first stem loop at the 5′end (Figure 3A). The antiterminator is much more stable than the terminator fold (ΔG = −22.9 Kcal/mol and −10.4 Kcal/mol, respectively), which suggests that the antiterminator should be the predominant conformation (Figure 3A). Considering that Rli51 mediates PTT in *L. monocytogenes* during exponential growth in BHI broth (Figure 2), these findings suggest that Rli51 adopts a terminator fold under these growth conditions.

Given that the terminator represents a less energetically favorable conformation than the antiterminator, we aimed to investigate the mechanism by which the terminator fold is stabilized under active growth in nutrient-rich conditions. The transition between two alternative RNA conformations (or conformational switching) can be triggered by the binding of a *trans*-acting factor (i.e., a small molecule, an sRNA or a protein) to a single-stranded RNA (ssRNA) segment [24,26,27]. Therefore, we first sought to determine if any of the ssRNA loops of Rli51 were essential to bring about PTT (Figure 3A, green and red font type). For this purpose, we created a series of translational fusions of *rli51* and *mpl-egfp* under the control of the *rli51* promoter. The constructs included *rli51* mutant versions with deletions in each of these ssRNA loops (Figure 3B, top panel). Elimination of the predicted terminator of Rli51 (Figure 3A, blue font) allowed for a complete Rli51-*mpl* transcriptional read-through (Figure 3B, *Δter*). The fact that deleting the predicted Rli51 terminator abolished PTT experimentally endorses the notion that Rli51 actually folds as a terminator when growing in BHI. In contrast, deletion of the 30-nucleotide-long segment that links Rli51 to *mpl* had no effect on Rli51 read-through into *mpl* (Figure 3B, Rli51 *), which is consistent with our previous observation (Figure 2B). Deletion of ssRNA loops 1 and 2 resulted in a 50 and 100% transcriptional read-through, respectively (Figure 3B, ΔL1 and ΔL2), indicating that these regions are important to mediate PTT in bacteria grown in BHI. Interestingly, the loss of PTT could not be attributed to a polar effect on Rli51 conformation resulting from L1 and L2 deletions because, in both constructs, the predicted terminator was not distorted (Appendix A). These results are consistent with the participation of ssRNA regions L1 and L2 as binding sites of cellular *trans*-acting factors to regulate PTT in exponentially growing bacteria.

We hypothesized that Rli51-mediated PTT could be triggered by the binding of regulatory sRNAs to L1 and/or L2 regions. To test this hypothesis, we first searched for sRNA regulators that potentially bind L1 or L2. We used IntaRNA [28] to predict interactions between Rli51 sRNA and all *L. monocyotgenes* EGD-e genes annotated as sRNAs in the Listeriomics database [29]. We carried out this analysis with the wild-type 121-nucleotide-long *rli51* (WT) and the ΔL1 and ΔL2 mutant sequences. From the output results, we first performed a selection of the most reliable predicted interactions. To do so, we selected the top 25% predicted interactions obtained with Rli51 wild-type sequence—based on the interaction energy (ΔG)—that mapped to Rli51 regions L1 or L2. Then, we computed the difference in the ΔG in the ΔL1 and ΔL2 mutant sequences in this group of selected predicted interactions and ranked them based on this difference (Figure 3C; Appendix A). Using this approach, we identified that Rli54, Rli46, and Rli21 potentially bind the L2 region, Rli113, and Rli126 L1. Additionally, Rli123 was identified to have binding sites in both the L1 and L2 regions (Figure 3C).

To experimentally verify if the identified sRNAs interact with Rli51 in vivo*,* we performed an MS2 affinity purification (MAP) of Rli51-binding RNAs (Figure 3D, left panel). MAP is an RNA pull-down approach in living bacterial cells that relies on the high-affinity and specific interaction between the coat protein (CP) of the MS2 phage and a hairpin aptamer (MS2) from a viral RNA genome [30]. To accomplish this aim, we generated a chimeric construct consisting of two copies of the MS2 aptamer fused to *rli51,* which was cloned downstream with an inducible *tet* promoter and transformed into a Δ*rli51* background. The hybrid RNA (MS2-Rli51) was induced with anhydrotetracycline, and the bacterial lysates were subsequently incubated with a protein composed of the CP fused to the maltose-binding protein (MBP::MS2-CP). The MBP serves as an affinity tag to purify MS2-Rli51 together with its binders (Figure 3D, left panel) [31]. To assess whether the candidate sRNAs identified in silico actually bind Rli51, we analyzed their abundances in the pull-down fractions obtained from the strain expressing MS2-Rli51 and from a control without the MS2 hairpins (Rli51) (Figure 3D). Rli51 sRNA showed more than a 2-log_10_ enrichment in MS2-Rli51 pull-down fractions compared to those obtained with the Rli51-expressing (without MS2 hairpins) control strain (Figure 3D, left Y-axis). Among the predicted Rli51-binding partners, Rli21, which was predicted to bind to L2 in Rli51 (Figure 3C; Appendix A), showed a six-fold enrichment in the MS2-Rli51 pull down compared to the control Rli51 pull down (Figure 3D, right Y-axis). Surprisingly, although Rli123 was predicted to have an extensive binding, spanning from L1 to L2, it did not show a clear enrichment in MS2-Rli51 pull down (Figure 3D, right axis). These results experimentally verified that Rli21 binds to Rli51 in exponentially growing *L. monocytogenes*. Overall, these results suggest that *rli51-mpl* PTT under active growth in laboratory conditions results from the stabilization of a transcriptional terminator triggered by the binding of a cellular *trans*-acting factor, such as Rli21.

### 2.4. Rli51 Is a Trans-Acting sRNA in Bacteria Infecting Epithelial Cells

We have observed that the induction of the *rli51* promoter in intracellular *L. monocytogenes* releases Rli51-mediated PTT, allowing for transcriptional read-through into *mpl* (Figure 2A,C). Nevertheless, more than 50% of total Rli51 is still represented by the short 121-nucleotide-long sRNA version (Figure 2A). We wondered about the physiological role of Rli51 sRNA and hypothesized that it could function as a *trans*-acting sRNA in intracellular bacteria. To test this hypothesis, we sought to compare the transcriptome of wild-type EGD-e and Δ*rli51* in intracellular bacteria. To accomplish this aim, we extracted total RNA from bacteria collected from infected human epithelial JEG3 cells and performed RNAseq (Figure 4A). A total of 3447 features were detected in both EGD-e and ∆*rli51* samples, from which 225 were significantly upregulated and 143 downregulated in intracellular ∆*rli51* (*p* < 0.05; Appendix A and Appendix A). Principal component analysis allowed us to discriminate between wild-type and ∆*rli51* samples on the basis of one single dimension, which captured 87.8% of the total variation (Appendix A). 

Functional enrichment analysis identified GO terms related to nucleotide binding and metabolism for upregulated genes, while for the downregulated genes, the most significant terms were associated with translation, ribosomes, and response to iron (Appendix A and Appendix A). Consistently, analysis of the protein–protein interaction networks of the differentially expressed genes (DEGs) using the STRING database showed clustering with high confidence (STRING score higher than 0.700) (Appendix A and Appendix A). Thus, the loss of *rli51* sRNA expression in intracellular bacteria results in a broad reshaping of the *L. monocytogenes* transcriptome, which suggests that the short sRNA version of Rli51 functions as *trans*-acting a regulator in this condition. 

Comparative transcriptomics of intracellular ∆*rli51* and EGD-e showed a consistent impact on gene expression; however, this alone is not enough to prove that Rli51 functions as a *trans*-acting post-transcriptional regulator. Direct regulation by *trans*-acting sRNAs entails imperfect base pairing with target mRNAs; therefore, we next sought to demonstrate that Rli51 interacts with target candidates. We used intaRNA to predict binding between Rli51 and the top DEGs (FDR < 0.05) in intracellular Δ*rli51* compared to EGD-e (Figure 4A,B; Appendix A). The analysis predicted extended interactions for the three most downregulated genes in intracellular Δ*rli51* (*rli38*, *lmo2186,* and *lmo0937*), and for the upregulated *lmo0207* (*orfZ*). In addition, the predicted binding sites in Rli51 for these targets lay in critical ssRNA loops L1, L2, or both (Figure 4B; Appendix A). Interaction energies were quite varied, with the strongest interactions found in the L1 and/or L2 regions of Rli51 (Appendix A), suggesting a role for these regions also in *trans*-mediated regulation. Interestingly, according to the prediction, Rli51 could bind *lmo0937* and *lmo2186* on more than one binding site, suggesting that the net interaction energy might be even more favorable (Appendix A). 

Finally, we carried out an in vivo Rli51 pull-down assay to verify the predicted interactions of Rli51 with target candidates experimentally. We selected the top three target candidates based on the magnitude of the fold expression change in the comparative intracellular transcriptomics of Δ*rli51* and EGD-e: Rli38, *lmo0937,* and *lmo2186* (Figure 4A). MS2 affinity purification (MAP) followed by qPCR quantification indicated a specific enrichment of *lmo0937* and *lmo2186* RNAs in MS2-Rli51 pull downs compared to Rli51 control samples, while Rli38 showed a negligible change (Figure 4C). These results indicate that Rli51 interacts with *lmo0937* and *lmo2186* in vivo. The difference in the enrichment between *lmo0937* and *lmo2186* (Figure 4C) might be an indication of the stability of the interaction, which depends on multiple factors other than the predicted interaction energy. Overall, these results verify that Rli51 binds cellular RNAs whose expression is altered in the absence of Rli51, which proves that the short version of Rli51 sRNA functions as a *trans*-acting sRNA in intracellular bacteria.

## 3. Discussion

Rli51 is an sRNA specific from pathogenic *Listeria* species induced under infection-relevant conditions [7,8,9,23,32,33]. However, both the mechanism of action and the physiological role of Rli51 have remained largely unexplored. Here, we have found that although Rli51 belongs to the 5′UTR of *mpl* mRNA, it does not play any role in the upregulation of *mpl* under infection conditions. Instead, Rli51 mediates a premature termination of transcription (PTT) in bacteria actively growing in a rich medium at 37 °C, which results in the production of a short 121-nucleotide-long sRNA (Figure 5). Under these conditions, the transcriptional activity of the promoter is very low, which could allow Rli51 to fold as a transcriptional terminator with the assistance of *trans*-acting sRNAs, such as Rli21/RliI (Figure 5). By contrast, PrfA-dependent induction of *rli51* in intracellular bacteria leads to an increased abundance of short Rli51 while, at the same time, enables a transcriptional read-through into *mpl* (Figure 5). Consequently, under infection-relevant conditions, Rli51 occurs in two similarly abundant RNA species: a 121-nucleotide-long sRNA and the 5′UTR of *mpl* mRNA. The accumulation of short Rli51 sRNA allows it to regulate target transcripts in intracellular bacteria, such as Lmo2186 and Lmo0937 (Figure 5). Thus, Rli51 can regulate gene expression through two different mechanisms: (i) as a *cis*-acting sRNA, where Rli51 would contribute to preventing leaky *mpl* transcription under non-infection conditions by mediating a conditional PTT, and (ii) as a *trans*-acting sRNA, where increased Rli51 levels in intracellular bacteria would contribute to fine tuning a pathogen’s response to the environment encountered within eukaryotic cell compartments.

RNA-mediated post-transcriptional control of genes involved in virulence is a common regulatory mechanism in bacterial pathogens [4,34], and in *L. monocytogenes* in particular [35]. Essential virulence factors in *L. monocytogenes* are post-transcriptionally regulated through their 5′UTRs [17]. For instance, the 5′UTR of *hly*, *actA,* and *inlA* is necessary to achieve maximal expression levels under infection-relevant conditions [15,20,36]. By contrast, in the case of the 5′UTR of *mpl*, i.e., Rli51, it functions as a transcriptional attenuator and does not have a relevant role in the induction of *mpl* in intracellular bacteria. The 5′UTR of *prfA* is also a negative regulator that acts as a temperature-sensing regulator, which allows for PrfA translation at 37 °C (the temperature of the host) while preventing it at lower temperatures [37]. In addition, the S-adenosylmethionine riboswitches in *L. monocytogenes* can release the sRNAs SreA and SreB, which, in turn, regulate *prfA* by binding to its 5’UTR, suggesting a link between nutrient availability and virulence [26]. Similarly, it would be interesting to identify the physiological signals that ultimately control Rli51 function and, thus, Mpl. 

Transcriptional attenuation refers to a blockade imposed by a specific conformation of an RNA segment (referred to as terminator) that functions conditionally. In a particular condition, binding of *trans*-acting factors (such as the ribosome, a small molecule, an RNA-binding protein, or an sRNA) can stabilize the formation of a terminator hairpin that impedes transcriptional read-through into the downstream gene [38]. In addition, in Gram-negative bacteria, the Rho factor serves as a global attenuator that terminates transcription of specific genes conditionally; sRNA binding within 5′ UTR of those genes disrupts Rho-mediated termination, thus leading to transcriptional read-through [39]. Therefore, sRNA-mediated antitermination is a common mechanism of bacterial gene regulation. We show that Rli51 is predicted to fold as a transcriptional terminator and as an antiterminator; the latter being the most stable conformation. We experimentally verified that the sRNA Rli21 (RliI) binds Rli51, and the predicted binding site lies in loop L2, a single-stranded RNA region that is critical for Rli51 to mediate PTT. This suggests that Rli21 binding stabilizes the formation of the terminator fold in Rli51. Interestingly, Rli21/RliI expression does not change under infection-relevant conditions compared to growth in nutrient-rich broth [8]. Therefore, in conditions of low transcriptional activity of the Rli51 promoter, Rli21/RliI might saturate Rli51 molecules, while under Rli51-inducing conditions, the excess free Rli51 would fold as an antiterminator (Figure 5). Such a regulatory scheme exerted by Rli21, and/or additional *trans*-acting factors, could explain the co-existence of both Rli51 and Rli51-*mpl* in intracellular bacteria. 

Rli51 mediates condition-dependent PTT, producing a 121 nt long sRNA that acts as a *trans*-acting regulator for other genes. This conclusion is supported by three observations: (i) the absence of Rli51 in intracellular Δ*rli51* bacteria altered the expression of a set of genes; (ii) we predicted binding of some of these candidate target transcripts to Rli51 within ssRNA loops L1 and L2; and (iii) we experimentally confirmed they bind Rli51 in vivo. Specifically, we verified that Rli51 binds and positively regulates the expression of Lmo2186. *lmo2186* is transcribed as a bicistronic mRNA together with *lmo2185. lmo2185* mRNA levels (Appendix A) are decreased in intracellular Δ*rli51* compared to EGD-e, suggesting that *lmo2185* and *lmo2186* are co-regulated by Rli51.

Lmo2185 and Lmo2186 were functionally annotated with the enriched GO term “response to iron” in our comparative transcriptomics (Appendix A). *lmo2185* and *lmo2186* encode the fur-regulated hemin-binding proteins Hbp1 and Hbp2, respectively, which are homologs of IsdA and IsdC in *Staphylococcus aureus* and function as a hemophore that scavenges ferric heme from the environment [40]. The IsdG family heme monooxygenase Lmo0484, which breaks down heme to release free iron [41], was also downregulated in intracellular Δ*rli51*. Additionally, *lmo0541* (*fhud*) and *lmo1960* (*fhuc*), encoding two subunits of the iron siderophore ABC transporter FhuBCDG, were also downregulated in Δ*rli51* (Appendix A). These findings suggest that Rli51 plays a role in upregulating iron scavenging systems under iron-limiting conditions, such as those found in intracellular compartments during infection. Interestingly, envelope stress induced by excess heme causes the LhrC4 sRNA to bind to *lmo2185*, *lmo2186,* and *lmo0484* transcripts, and, at least for *lmo0484*, this resulted in target transcript downregulation [42]. These results suggest that genes involved in heme uptake and degradation are post-transcriptionally regulated in response to both the excess and limitation of heme.

Our comparative transcriptomics revealed a consistent upregulation of genes belonging to LIPI-1: *hly, mpl, actA, plcB*, *orfX,* and *orfZ* (see *lmo0202* to *lmo0207* in Appendix A). *mpl, actA,* and *plcB* belong to the lecithinase operon and can be transcribed from the *rli51* promoter as a single mRNA [14]. Consistently, the deletion of *rli51* resulted in upregulated endogenous *mpl* or *mpl::gfp* translational fusions in intracellular bacteria, as determined by targeted transcript quantification by RT-qPCR (Figure 1A,C). Together, these results also suggest a *cis*-acting mechanism of regulation by Rli51 in intracellular bacteria. The transcriptional activity of the promoter is very high, so it seems that the impact of this negative regulation mechanism is limited during intracellular infection. On the other hand, *hly*, *orfX,* and *orfZ* are transcribed as monocistrons, which suggests that Rli51 could also regulate the expression of other virulence factors through a *trans*-acting mechanism. In fact, *orfZ* was the target candidate with the highest predicted interaction energy (Figure 4A and Appendix A). Future studies will help to ascertain if Rli51 has any role as a general repressor of the virulence program of *L. monocytogenes*. 

Our study sheds light on the function and mechanism of regulation mediated by an sRNA from LIPI-1 but entails limitations. For example, although we have identified Rli21/RliI as an sRNA that binds Rli51 at loop L2, our results do not preclude the involvement of additional sRNAs or other *trans*-acting factors (such as a small molecule) in the regulation of Rli51-mediated PTT. On the other hand, the deletion of a single sRNA typically results in hard-to-detect subtle phenotypes with modest expression changes in DEGs [43]. Our intracellular comparative transcriptomics was not meant to analyze comprehensively Rli51 regulon in intracellular bacteria though, for which other approaches might have shown clearer expression changes in putative target transcripts [5,44]. The aim of [5,44] was to identify physiologically relevant Rli51 targets in intracellular bacteria so as to prove a *trans*-acting mechanism. Further studies will help to determine a more detailed view of the function of Rli51 in virulence. 

Our work provides compelling evidence of the dual regulatory mechanism of Rli51, acting both in *cis* and in *trans*. Similarly, other sRNAs exhibit *cis*- and *trans*-acting regulatory mechanisms, influencing virulence [26,45], metabolism [46,47,48], and nutrient uptake [49], which suggests that this dual mechanism might be more common than previously anticipated [50]. In fact, global mapping of 3′ ends in different bacterial species revealed that pervasive conditional PTT populates the transcriptome with 5′UTR-derived sRNAs that might have regulatory functions [18,50,51]. As conditional PTT most often relies on riboswitches, these findings open a yet unexplored level of regulation that links the concentration of particular intracellular metabolites with the release of sRNA regulators [26], suggesting an exciting intracellular sensing and regulatory crosstalk to control bacterial homeostasis.

## 4. Materials and Methods

### 4.1. Bacterial Strains and Growth Conditions

The *Listeria monocytogenes* (*L. monocytogenes*) strain used here was isogenic to wild-type strain EGD-e (serotype 1/2a) [52]. The *rli51*-null mutant was generated by allelic replacement. Homology regions flanking the *rli51* locus were generated by PCR and cloned one after another into the thermo-sensitive suicide integrative vector pMAD between the BamHI and KpnI restriction sites. Primer pairs used for the amplifications were pMAD-BamHI-AB-Fw with AB-StuI-CD-Rv and AB-StuI-CD-Fw with CD-KpnI-pMAD-Rv. Integration, excision, and screening steps were performed as described [53]*. rli51* deletion was confirmed by Sanger sequencing. 

Unless otherwise stated, *L. monocytogenes* strains were grown at 37 °C in brain heart infusion (BHI) broth without shaking. To prepare the cultures in the exponential phase of growth in gene expression analysis experiments, bacteria were allowed to reach an OD_600_ = 0.2 and then diluted two consecutive times. Specifically, a single colony was inoculated in 10 mL of BHI and grown at 37 °C in standing conditions. When it reached an OD_600_ = 0.2, bacteria were diluted with fresh pre-warmed BHI and grown for at least 10 generations such that an OD_600_ = 0.2 is not exceeded. For inducible expression experiments, strains carrying constructs cloned in the tetracycline-inducible expression vector pRMC2 [54] were grown to the exponential phase (OD_600_ = 0.2) (see above) and then exposed to anhydrotetracycline (final concentration: 0.05 µM or 0.1 µM) for 40 min. To transform plasmids into *L. monocytogenes*, 1 µg of plasmidic DNA was electroporated at 2.4 kV, 200 ohms, and 25 μF in electrocompetent EGD-e and selected on plates supplemented with 10 µg/mL chloramphenicol (pRMC2 derivatives) or 5 µg/mL erythromycin (pMAD derivatives).

### 4.2. Cloning Procedures

Cloning details for each specific construct made for this work are included in Appendix A. Cloning procedures were performed in the DH5α *E. coli* strain. Bacteria were routinely grown in Luria Bertani (LB) broth at 37 °C with shaking at 180 rpms, and strains bearing pRMC2 or pMAD derivatives were selected with 100 µg/mL ampicillin. Ligation reactions and pure plasmids were transformed into chemically compentent bacteria [53]. Q5 DNA polymerase (M0491S), T4 DNA ligase (M0202S), and restriction enzymes (KpnI: R3142S, SalI: R3138S, EcoRI: R3101S, BglII: R0144S, EcoRV: R0195S, DpnI: R0176S) were supplied by New England Biolabs. Kits for extracting plasmidic DNA (NucleoSpin Plasmid: 740588.50) and DNA from agarose gels (NucleoSpin Gel and PCR Clean-up: 740609.50) were supplied by Macherey Nagel. Rli51and MS2::Rli51 constructs and GFP translational fusions were cloned into pRMC2 plasmid [54]. The GFP coding sequence was obtained from pAD1-cGFP [55]. All constructs were assessed by Sanger sequencing.

### 4.3. Determination of L. monocytogenes Intracellular Proliferation (Gentamicin-Resistance Assay) 

Human placental choriocarcinoma JEG-3 epithelial cells (ATCC HTB-36) and NRK-49F-9F (ATCC CRL-1570) rat fibroblasts were propagated in MEM and DMEM, respectively, supplemented with 10% FBS, 4 mM L-glutamine, and nonessential amino acids. Infections were carried on cells at an 80% confluence seeded on 24-well culture dishes. Prior to infection, *L. monocytogenes* strains were grown in BHI medium for 14 h at 37 °C in standing conditions. Bacteria were collected by centrifugation at 800× *g* for 1 min at room temperature and then resuspended in the corresponding cell culture medium. Bacterial suspension was added to the eukaryotic cells medium at a multiplicity of infection (MOI) of 10:1 (bacteria/eukaryotic cell). After a 60 min long co-incubation, non-internalized bacteria were removed by three consecutive washing steps with pre-warmed Hank’s Balanced Salt Solution (HBSS) (Gibco). To kill the remaining extracellular bacteria, a fresh medium containing 10 µg/mL gentamicin was added to infected cells. After incubation for 2 or 6 h in these conditions, the eukaryotic cells were lysed with 100 µL of a solution containing 1%TritonX-100–0.1%SDS for 5 min at room temperature. Serial dilutions of this suspension were spread onto BHI agar plates, and the colony-forming units (CFUs) were counted after overnight incubation at 37 °C. The intracellular proliferation rate was calculated by dividing the number of CFUs at 6 h post-infection by that at 2 h post-infection.

### 4.4. RNA Isolation from Cultured and Intracellular Bacteria 

To prepare the samples for RNA extraction from bacteria grown in BHI broth (supplemented with 10 µg/mL chloramphenicol in pRMC2 plasmid-bearing strains), 0.2 volumes of an ice-cold solution containing 5% phenol and 95% ethanol were added to the culture and then kept for 30 min in ice to quench RNA metabolism. The suspension was scrapped off the plate and the bacteria collected by centrifugation at 9300× *g* at 4 °C for 10 min. Afterwards, the bacterial pellet was washed with 1 mL of an ice-cold solution containing 1% phenol and 19% ethanol diluted in 0.1% DEPC-treated water, transferred into a 1.5 mL tube, re-centrifuged at 9600× *g* for 5 min, and stored at −80 °C until used.

To obtain intracellular bacteria, human epithelial JEG-3 cells (ATCC HTB-36) were cultured with MEM supplemented with 10% FBS, 4 mM L-glutamine, and nonessential amino acids on 500- cm^2^-BioDish-XL plates (351040, BD Biosciences, Franklin Lakes, NJ, USA) to an ∼80% confluence (∼5.6 × 10^7^ cells per plate). A single colony was inoculated in 1.5 mL of BHI (supplemented with 10 µg/mL chloramphenicol in pRMC2 plasmid-bearing strains) in a 10 mL glass tube with a vented cap and grown overnight at 37 °C without shaking. This culture was spun down for 1 min at 800× *g* at room temperature and resuspended in a serum-free culture medium at the same cell density. For each intracellular bacterial sample, two 500 cm^2^ plates were infected for 30 min at an MOI of 10:1 (bacteria/eukaryotic cell). Non-internalized bacteria were washed away with pre-warmed HBSS and killed by adding a fresh culture medium containing 100 µg/mL gentamicin. After a 2-h incubation, the infected cell culture was washed twice with pre-warmed HBSS, and a fresh medium supplemented with 10 µg/mL gentamycin was added. At 6 h after infection, the eukaryotic cell culture was washed five times with HBSS and lysed in a chilled solution containing 0.1% SDS, 1% acidic phenol, and 19% ethanol diluted in 0.1% DEPC-treated water for 30 min at 4 °C. The eukaryotic cell lysate was scrapped off the plate, and the intracellular bacteria were collected by centrifugation (27,500× *g*, 4 °C, 30 min). Bacteria were washed with the same lysis solution and re-pelleted (15,500× *g*, 4 °C, 10 min), and the pellet was stored at −80 °C.

Frozen pellets from bacteria cultured in BHI (total amount 3 ODs: 15 mL at an OD_600_ = 0.2) and from intracellular bacteria (obtained from two 500 cm^2^ plates of infected JEG-3 cells) were thawed, resuspended in a freshly prepared solution containing 10% glucose, 12.5 mM Tris-HCl pH 7.6, and 65 mM EDTA. Resuspended cells were transferred to a tube containing 1 volume of acidic phenol. This cell suspension was beaten with acid-washed glass beads (0.4 g ≈ ca. 0.2 mL) twice for 45-s-long cycles at 6 m/s in a FastPrep 24 (MP Biomedicals). As a centrifugation step to pellet unsoluble material and separate the phenolic phase (14,000× *g*, 4 °C, 10 min), RNA was extracted from the aqueous phase with 1 mL of NZYzol (MB18501, Nzytech, Lisboa, Portugal) followed by two consecutive extraction steps with 0.5 mL of chloroform and then precipitated with isopropanol for 30 min at room temperature. RNA yield was quantified by spectrophotometry with a nanodrop ND2000 (Thermo Fisher Sci., Waltham, MA, USA), and 5 µg of RNA was treated with 3 U of DNase I (Turbo DNA-free kit: AM1907, Thermo Fisher Sci.) for 60 min at 37 °C. RNA integrity was assessed by running a 200 ng sample of the extracted RNA through agarose TAE electrophoresis. Genomic DNA carryover contamination-free RNA was verified by a negative result in a 30-cycle PCR using rpoB-F and rpoB-R primers (Appendix A) and 20 ng of each RNA sample.

### 4.5. Reverse Transcription and Real-Time Quantitative PCR (qPCR)

cDNA samples were prepared from 1 µg of total DNA-free RNA using the Applied Biosystems High Capacity cDNA Reverse Transcription Kit (4368814, Thermo Fisher Sci.) including a mix of random hexamers. The RT-PCR protocol was 10 min at 25 °C, 120 min at 37 °C, and 5 min at 85 °C. qPCR was performed in a 10 µL final reaction volume, including 1 ng of cDNA as a template, 500 nM of each target gene-specific primer (Appendix A), and 5 µL of the Applied Biosystems Power SYBR Green PCR Master Mix (4367659, Thermo Fisher Sci.). qPCR runs were performed in an Applied Biosystems 7500 Real-Time PCR System instrument (Thermo Fisher Sci.) using a standard amplification protocol (10 min at 95 °C; 45 cycles of 15 s at 95 °C and 1 min at 60 °C; dissociation curve of 15 s at 95 °C, 1 min at 60 °C, and a progressive temperature increase until 95 °C). Target-specific primers (listed in Appendix A) were designed using Applied Biosystems Primer Express 3.0 (Thermo Fisher Sci.). As a quality control of each primer pair, the amplification efficiency of the corresponding qPCR assay was determined by running a standard curve comprising 12.5 ng of a pool of cDNA samples and five consecutive 1/5 serial dilution points. For each sample, a reverse transcription-minus control including 80 ng of RNA was analyzed with the same primer set to assess the extent of genomic DNA contamination.

For absolute RNA quantification, every plate included a standard curve made up of 5 points represented by serial dilutions of an EGD-e genomic DNA sample (12.5, 2.5, 0.5, 0.1, and 0.02 ng of gDNA). To calculate the number of molecules of target genes in each point of the standard curve, we considered a molecular weight of EGD-e gDNA of 1.82 × 10^9^ g/mol (gDNA length 2.94 × 10^6^ bps, average MW per nucleotide in dsDNA 619.96 g/mol) and the Avogadro No. 6.023 × 10^23^ molecules/mol. The fluorescence threshold was set such that it crossed the amplification curves (cycle vs. ΔRn) of all concentration points in the standard curve within the linear region of the amplification plot (exponential phase). Each experimental cDNA sample was run in triplicate, and the corresponding Ct was calculated as the mean of the three determinations. Individual technical replicates that deviated more than 0.75 cycles from the other two ones were not considered to calculate the mean Ct value. The No. of molecules of target mRNA transcripts in each experimental sample was calculated by interpolation of the corresponding mean-Ct values in the standard curve plot (Log_10_ [No. of molecules] vs. Ct). Absolute quantification of each target mRNA transcript was expressed as the No. of molecules per nanogram of total extracted RNA in each sample. The relative quantification of target mRNAs was calculated by dividing its absolute quantification by that of the *rpoB* transcript. Enrichment ratios in RNA pull-down experiments were calculated by dividing the relative quantification of each specific RNA in MS2-Rli51 eluate by that in the eluates of Rli51 control pull downs. *rpoB* was used as a housekeeping gene. Every qPCR assay included a non-template control (NTC), which included the same volume of H_2_O instead of the cDNA template.

### 4.6. Production of Recombinant His-MBP-CP

A 500 mL culture of a BL21 *E. coli* strain with pMAL-C2::His-MBP-CP [56] was grown to OD_600_ = 0.6 in LB supplemented with 10 mM glucose and 100 µg/mL ampicillin (final volume = 500 mL). His-MBP-CP ectopic expression was induced for 2 h after the addition of IPTG (final concentration: 0.3 mM). Afterwards, bacteria were collected by centrifugation (4000× *g*, 4 °C, 20 min), and the pellet was frozen for 24 h at −20 °C. Bacteria were thawed, resuspended in 10 mL of a lysis buffer (50 mM Na_2_HPO_4_-NaH_2_PO_4_ pH = 8, 300 mM NaCl, 1% triton X-100, 10 mM β-mercaptoethanol, 10% glycerol, and 1× Roche EDTA-free complete protease inhibitor cocktail [04693116001, Merck, Rahway, NJ, USA]) per gram of bacterial wet weight and run through four consecutive freeze and thaw cycles in a dry ice/ethanol bath. The suspension was then sonicated (15 pulses of 15 s with 50% amplitude with 1-min-long breaks in ice in between) in a Q500 sonicator with a CL334 converter (QSonica, Newtown, CT, USA), and the lysate was clarified by centrifugation (9000× *g*, 4 °C, 30 min). To purify His-MBP-CP, 3 mL (bed volume) of TALON^®^ Metal Affinity Resin (635503, Takara, San Jose, CA, USA) were equilibrated by incubating the resin in batch with 6 volumes of washing buffer (lysis buffer without glycerol and protease inhibitors) for 3 times. The equilibrated resin was incubated with the clarified lysate at a volume ratio of 1/10 (resin/lysate) on a wheel for 2 h at 4 °C, washed 6 times in batches with 6 volumes of washing buffer, and then eluted with washing buffer supplemented with 220 mM imidazole (12 fractions, 250 µL each). Protein concentration was determined with the Pierce BCA reagent (23225, Thermo Fisher Sci.), and the protein-containing fractions were pooled (ca. total volume: 1 mL) and dialyzed against 600 mL of RLN buffer (50 mM Na_2_HPO_4_-NaH_2_PO_4_ pH = 7.4, 140 mM NaCl, 1.5 mM MgCl_2_, 1 mM DTT) at 4 °C for 16 h.

### 4.7. MS2-Affinity Purification of Rli51-Binding RNAs (RNA Pull Down)

To prepare bacterial lysates, *L. monocytogenes* EGD-e strains Δ*rli51*:pRMC2-*rli51*-MS2 and Δ*rli51*:pRMC2-*Rli51* were grown to exponential phases (final volume of 600 mL, OD_600_ = 0.12) in BHI supplemented with 10 µg/mL chloramphenicol. Then, Rli51 and MS2-Rli51 expression was induced by adding anhydrotetracycline (ATc; final concentration: 0.2 µM). When the cultures reached an OD_600_ = 0.2 (ca. after a 40-min-long incubation with ATc), bacteria were collected by centrifugation (8500× *g*, 4 °C, 10 min), washed with column buffer (20 mM Tris-HCl pH 7.4, 200 mM NaCl, 1 mM EDTA, 1 mM DTT), and frozen at −80 °C for 24 h. The pellet was resuspended in 1 mL of column buffer, subjected to three consecutive freeze and thaw cycles in a dry ice/ethanol bath, and beaten with 0.4 g (ca. 0.2 mL) of acid-washed glass beads for three 20-s-long cycles at 2 m/s in a FastPrep 24 (MP Biomedicals, Auckland, New Zealand). After removing the beads, 4 µL of the Ambion™ RNase Inhibitor (cloned, 40 U/µL; AM2684, Thermo Fisher Sci.) were added to the lysate and then clarified by centrifugation (17,500× *g*, 4 °C, 30 min).

To prepare the affinity resin for MS2-tagged RNA pull-down experiments, the recombinant purified His-MBP-CP protein was first bound to amylose resin. A total of 500 µL (bed volume) of amylose resin (E8021S, New England Biolabs, Ipswich, MA, USA) was equilibrated in column buffer (6 resin volumes, 5 min, 3 times). This equilibrated resin sample was incubated in a batch with 10 volumes of column buffer containing 0.75 mg/mL of His-MBP-CP on the wheel (4 °C, 2 h). The resin was washed 4 times with 6 resin volumes of column buffer and stored at 4 °C until used (≤5 days).

To conduct the RNA pull down, a 100 µL aliquot (bed volume) of the amylose resin bound to His-MBP-CP (ca. 0.715 mg of His-MBP-CP) was mixed with 1 mL of the bacterial lysate and let to incubate for 2 h on the wheel at 4 °C. The resin was washed 4 times with 6 volumes of column buffer and subsequently transferred to a disposable empty Poly-Prep^®^ chromatography column (7311550, Bio-Rad, Hercules, CA, USA). Elution was performed with 3 volumes of the column buffer supplemented with 10 mM maltose. The fractions were pooled (total volume: 300 µL), and the RNA was extracted with NZYol (see above).

### 4.8. PASIFIC Structural Analysis of Rli51

In silico analysis to predict the occurrence of alternative terminator/antiterminator RNA conformation compatible with a riboswitch or an attenuator was carried out using the PASIFIC webserver (https://www.weizmann.ac.il/molgen/Sorek/PASIFIC/; accessed on 16 March 2022) [25]. A 128-nucleotide-long sequence starting from the transcriptional start site of rli51 was selected as the input. This included the 121 nucleotides annotated as the Rli51 transcript plus a U-rich 7-nucleotide-long downstream sequence (AGUUUUA). Default parameters optimized to detect attenuators were used.

### 4.9. sRNA-sRNA Interaction Prediction

In silico prediction of sRNA-Rli51 pairwise interactions was performed by conducting multiple one-to-one interaction analyses of each sRNA with Rli51 using the command-line version of IntaRNA 2.0 [28,57]. A batch analysis was run using a manually created *L. monocytogenes* sRNAs list file in FASTA format as the query and the Rli51 sequence as the target. sRNA sequences were obtained from the Listeriomics database [29].

### 4.10. RNA Sequencing: Library Construction and Next-Generation DNA Sequencing

Library construction and DNA sequencing were performed by Novogene. Prokaryotic RNA samples were depleted of ribosomal RNA and purified by ethanol precipitation. After fragmentation, the first strand of cDNA was synthesized using random hexamer primers. During the second strand cDNA synthesis, dUTPs were replaced with dTTPs in the reaction buffer. The directional library was ready after end repair, A-tailing, adapter ligation, size selection, USER enzyme digestion, amplification, and purification. The library was checked with Qubit and real-time PCR for quantification and a bioanalyzer for size distribution detection. Quantified libraries will be pooled and sequenced on the Illumina platform Novaseq6000, according to effective library concentration and data amount required.

### 4.11. RNA Sequencing Data Analysis

For data analysis, raw reads were curated by trimming the adapters and removing poly-N sequences and low-quality reads. Only high-quality data were used for downstream analysis. Bowtie2 software [58] was used for indexing the reference genome (NC_003210.1) and mapping the reads. Reads mapping to each gene were counted by FeatureCounts and used to calculate the FPKM for estimating the expression levels of each gene [59]. Principal component analysis (PCA) using these expression levels was conducted using the FactoMineR R package to determine the overall reproducibility of our RNAseq biological replicates. Differential expression analysis between EDG-e (control) and ∆*rli51* sample groups was performed by the DeSEQ2 R package based on the negative binomial distribution, and *p*-values were adjusted by the Benjamini–Hochberg correction to control the false discovery rate. Only those features with a *p*-value < 0.05 were considered significant. A heatmap of the normalized expression levels of each significant gene was visualized using Morpheus from the Broad Institute (https://software.broadinstitute.org/morpheus; accessed 20 November 2023). For normalization, each gene FPKM was normalized by the average of the FPKM of that gene in all samples and then transformed by log2. The ClusterProfiler R package was used to perform statistical enrichment of differential expression genes based on GO terms and known regulons of *L. monocytogenes* available in curated databases: PRODORIC (http://www.prodoric.de/; accessed 5 February 2021), RegPrecise (https://regprecise.lbl.gov/index.jsp; accessed 5 February 2021), and CollectTF (http://www.collectf.org/browse/home/; accessed 5 February 2021). Protein–protein interaction (ppi) networks were built by searching the differential expression genes in the STRING database (http://string-db.org/; accessed 20 November 2023).

## 5. Conclusions

Rli51 is an sRNA located in the 5′UTR of the LIPI-1 virulence factor *mpl* that functions both as a *cis*- and *trans*-acting sRNA.Under nutrient-rich laboratory conditions, the transcription of *rli51-mpl* is prematurely terminated, producing a 121-nucleotide-long sRNA at very low levels.The sRNA Rli21/RliI binds to a single-stranded RNA loop in Rli51, which is crucial for premature transcription termination.Under intracellular infection conditions, *rli51* is induced, leading to a higher abundance of the short Rli51 sRNA and allowing for transcriptional read-through into *mpl*.Transcriptional read-through allows Mpl virulence factor expression in intracellular bacteria and the short Rli51 sRNA regulates in *trans* transcripts encoding iron scavenging and cell surface proteins.

## Figures and Tables

**Figure 1 ijms-25-09380-f001:**
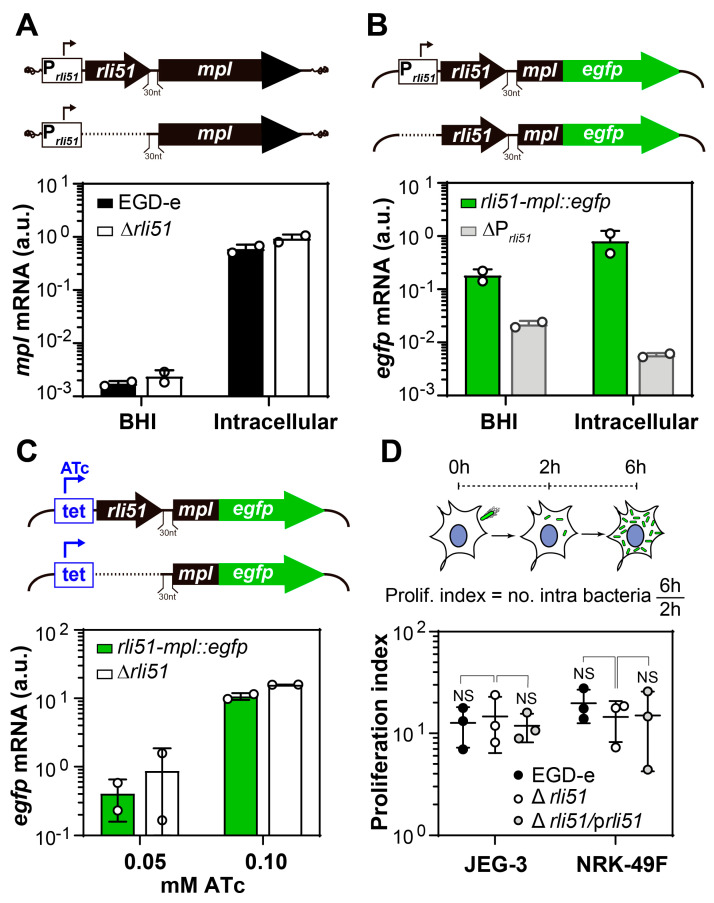
Rli51 does not exert a *cis*-acting regulation over *mpl* expression in intracellular bacteria. (**A**) *mpl* transcript quantification by RT-PCR in EGD-e (black bars) and Δ*rli51* strains (white bars) grown in BHI at 37 °C or collected from infected JEG-3 cells. (**B**) *mpl*-*egfp* transcript quantification in EGD-e transformed with plasmids expressing Rli51-*mpl*::*egfp* translational fusions with (green bars) or without the *rli51* promoter (gray bars, ΔP*_rli51_*) grown in BHI at 37 °C or collected from infected JEG-3 cells. (**C**) *mpl*-*egfp* transcript quantification in EGD-e transformed with plasmids expressing inducible *mpl*-*egfp* translational fusions with (green bars) or without Rli51 (Δ*rli51*) with different concentrations of the inductor anhydrotetracycline (ATc). Plotted in (**A**–**C**) are the number of molecules of *mpl* or *egfp* per nanogram of total extracted RNA divided by those of *rpoB* (used as a housekeeping gene). Bars and error bars represent the mean and standard deviation of two independent experiments. (**D**) Gentamicin-resistance assays in JEG-3 epithelial and NRK-49F fibroblast cells of EGD-e (black points), Δ*rli51* (white points), and Δ*rli51* complemented with an inducible plasmid-expressing Rli51 (gray points). Points represent the number of intracellular bacteria at 6 h divided by those at 2 h after infection (proliferation index). NS: non-significant in one-way ANOVA analysis.

**Figure 2 ijms-25-09380-f002:**
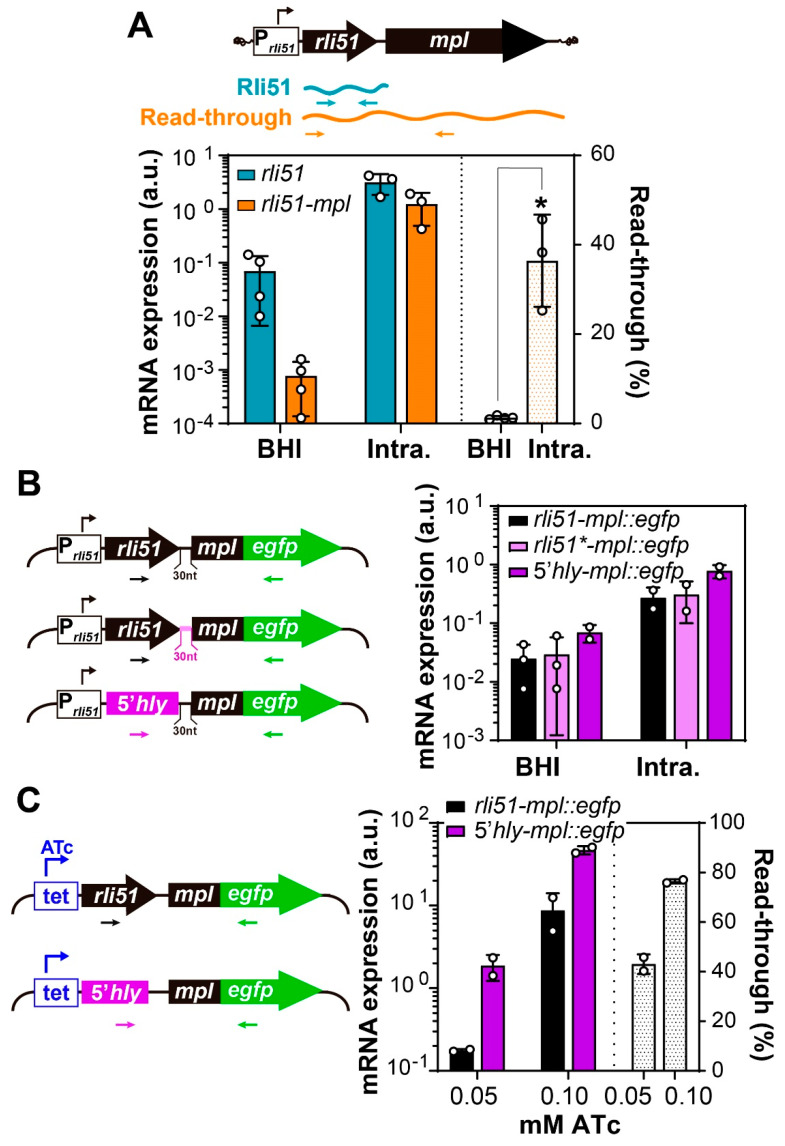
Rli51 mediates a premature termination of *rli51-mpl* transcription in *L. monocytogenes* growing in BHI. (**A**) Quantification of total Rli51 (Blue) and Rli51-*mpl* transcript (orange) in *L. monocytogenes* grown in BHI at 37 °C (BHI) and in bacteria collected from JEG-3 cells at 6 h after infection (Intra). Rli51-*mpl* read-through (dotted bars, right Y-axis) corresponds to the ratio between Rli51-*mpl* and total Rli51 (in molecules per nanogram of total cellular RNA) multiplied by 100. The asterisk denotes a significant difference (*p*-value < 0.05) as determined by unpaired Student’s t-test with Welch correction. (**B**) Quantification of *mpl* transcription coupled to its 5′UTR from translational fusions to *egfp* driven by the Rli51-*mpl* promoter (P_Rli51_) and includes (i) Rli51 (black), (ii) Rli51 without the 30 nt long sequence that links it to *mpl* (Rli51 *, violet)), and (iii) the 5′UTR of *hly* (purple). (**C**) Same as in (**B**) but with translational fusions driven by an inducible *tet* promoter. Different levels of coupled transcripts were obtained using 0.05 and 0.1 mM of the inducer anhydrotetracycline (ATc). Rli51-*mpl* read-through was calculated as in (**A**).

**Figure 3 ijms-25-09380-f003:**
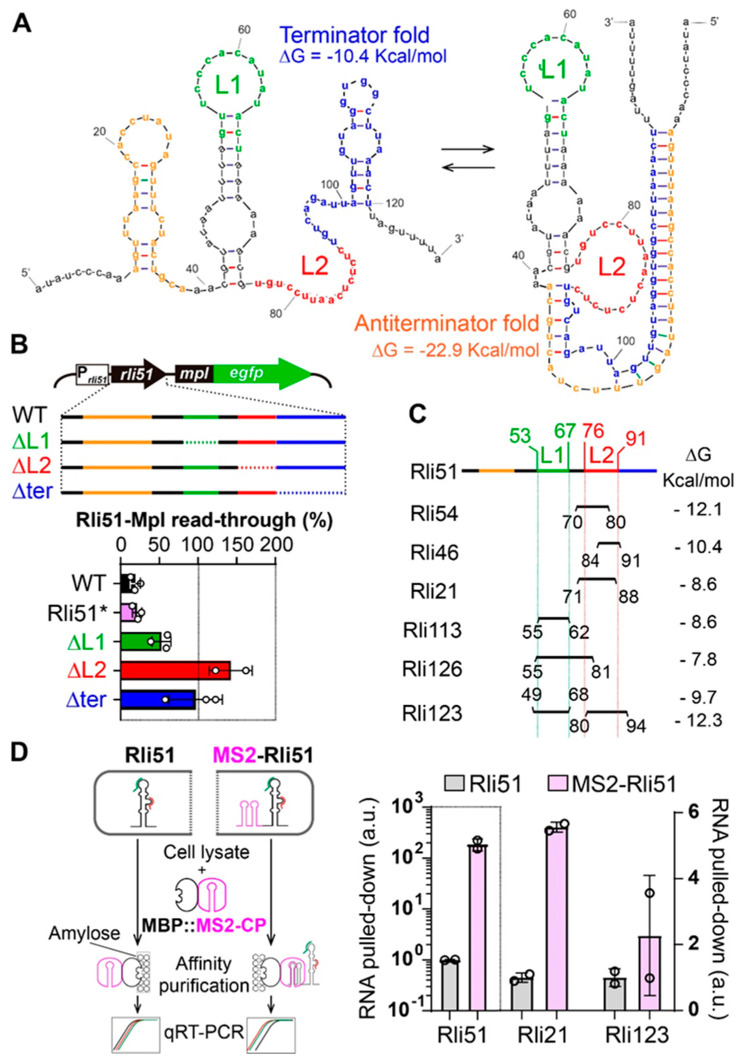
Rli51 is a condition-specific transcriptional attenuator of *mpl*. (**A**) Prediction of alternative terminator/antiterminator conformations of Rli51 compatible with an attenuator element using the PASIFIC webserver. Blue: predicted terminator. Green and red: single-stranded RNA (ssRNA) segments prevalent in both terminator and antiterminator folds. Orange: segment that hybridizes with the predicted terminator, thus allowing for transcriptional read-through. (**B**) Transcriptional read-through (calculated as in Figure 2A) in translational fusions to *egfp* with the *rli51-mpl* promoter (P_Rli51_). The constructs include *rli51* and *rli51* without the 30-nt long sequence that connects it to *mpl* (Rli51*; see Figure 2B), three deletion mutants of the ssRNA segments L1 (ΔL1, green) and L2 (ΔL2, red), and the predicted terminator (Δter, blue). The graph on the top outlines *rli51* sequence, where each colored segment represents the regions described in (**A**) and the dotted line represents the deleted region in each construct. (**C**) Prediction of sRNAs that potentially bind L1 and L2 ssRNA segments in Rli51 using intaRNA. (**D**) Rli51-binding RNAs pull down using MS2 affinity purification (MAP) coupled to RT-qPCR. Left panel: summary of the procedure. Bait hybrid (MS2-Rli51) and control (Rli51) RNAs are induced in different strains, which are then lysed and incubated with the fusion protein MBP::MS2-CP. Affinity purification of Rli51 and its binders through the binding of MBP to amylose resin. Quantification of Rli51 potential binders in eluates was carried out by RT-qPCR. Right panel: quantification of potential Rli51 binders. Number of RNA molecules per nanogram of total extracted RNA normalized by the amount of *rpoB*. The points represent the enrichment ratio of each specific sRNA in MS2-Rli51 vs. Rli51 control pull downs. MS2: RNA aptamer that binds the coat protein of MS2 phage; MBP::MS2-CP: maltose-binding protein fuse to the coat protein of the MS2 phage. The bars and error bars represent the mean and the standard deviation, respectively.

**Figure 4 ijms-25-09380-f004:**
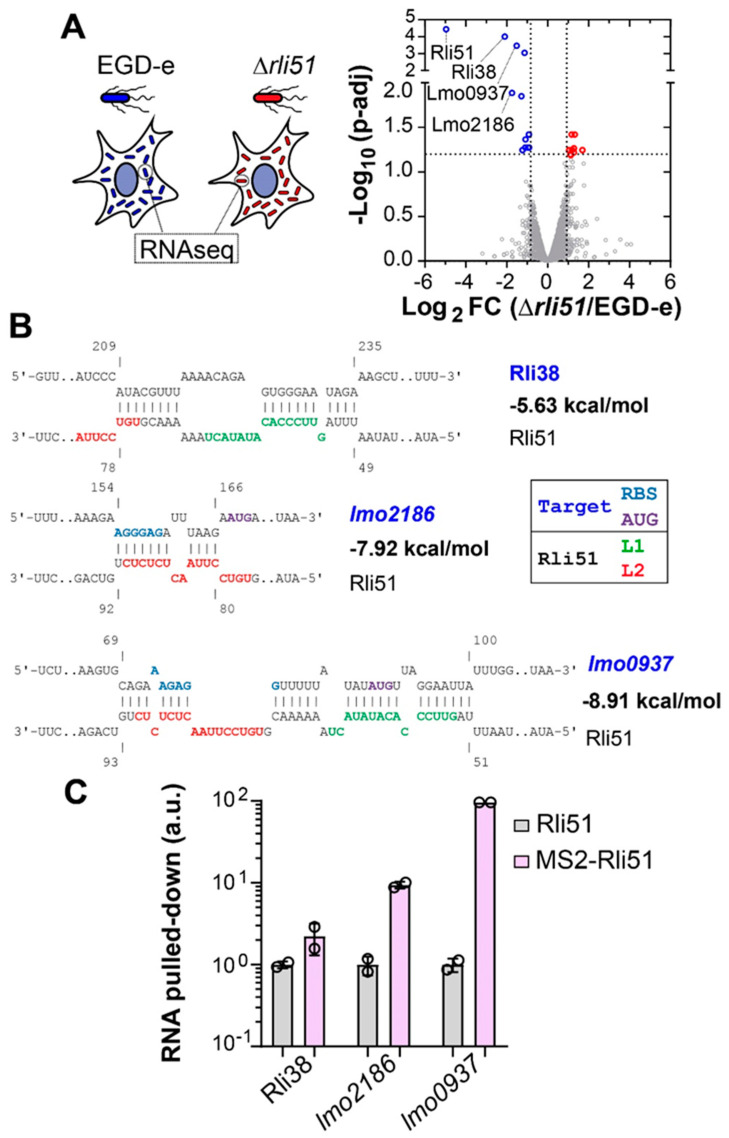
Rli51 functions as a *trans*-acting sRNA in intracellular bacteria. (**A**) Left panel. Comparative transcriptomics of Δ*rli51* and EGD-e *L. monocytogenes* strains collected from infected JEG-3 cells. Right panel. Volcano plot of comparative transcriptomics. X-axis represents differences in expression levels as the Log_2_-tranformed value of the fold change in each gene (ratio of the expression in Δ*rli51* and EGD-e). Y-axis represents statistical significance as the −Log_10_-transformed adjusted *p*-value calculated by the Benjamini–Hochberg correction of the false discovery rate in multiple comparisons. Dotted lines indicate thresholds used to select the top DEGs: p-adj < 0.05 and fold change ≥ 2. Red and blue points indicate the top down- and up-regulated genes in Δ*rli51*, respectively. (**B**) IntaRNA-based prediction of interactions between Rli51 and the top Rli51-regulated-gene candidates identified in comparative transcriptomics. Green and red indicate ssRNA segments L1 and L2 in Rli51, respectively. Blue and purple indicate the ribosome-binding site (RBS) and initiator codon AUG in the potential target, respectively. (**C**) Experimental verification of Rli51 binding to the top Rli51-regulated-gene candidates identified in comparative transcriptomics. Same as in Figure 3D.

**Figure 5 ijms-25-09380-f005:**
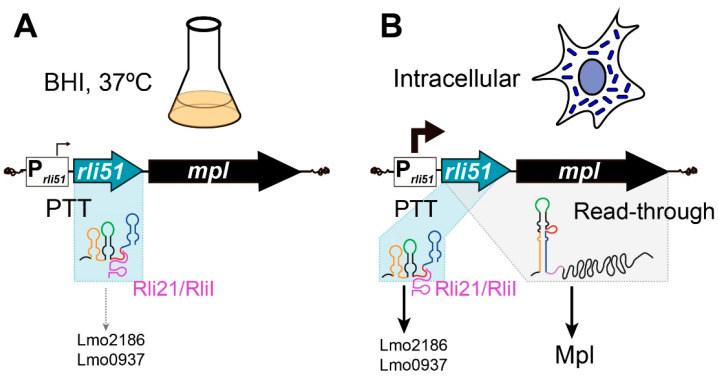
Model of the mechanisms of regulation of Rli51 sRNA. Rli51 is a transcriptional attenuator that mediates condition-specific PTT. (**A**) When bacteria are growing in nutrient-rich lab conditions (BHI, 37 °C), the transcriptional activity of its promoter is very low, and cellular trans-acting factors, such as Rli21/RliI, can bind to Rli51, which allows Rli51 to fold as a transcriptional terminator. This might prevent leaky expression of Mpl. (**B**) In infection-relevant conditions, such as in intracellular *L. monocytogenes*, PrfA increases the transcription from the Rli51 promoter. Increased expression of Rli51 might override Rli21/RliI-mediated Rli51 folding as a terminator termination, thus allowing transcription to proceed into *mpl*. Therefore, in intracellular bacteria, two Rli51 species are generated at high abundance: a short sRNA resulting from PTT (Rli51) and an mRNA made of Rli51 and *mpl* (Rli51-*Mpl*). Rli51 sRNA regulates *trans* target genes, such as *lmo0937* and *lmo2186*. On the other side, Rli-*mpl* can be translated, producing the metalloprotease Mpl.

## Data Availability

The datasets generated during and/or analyzed during the current study are available at the institutional repository of the Universidad Complutense de Madrid Docta Complutense (https://docta.ucm.es/home; accessed on 25 August 2024) and from the corresponding authors (ad.ortega@ucm.es; mgpuccia@cbm.csic.es) on reasonable request. Materials described in the manuscript will be freely available to any research to use them for noncommercial.

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
