# Peer review of "Rli51 Attenuates Transcription of the Listeria Pathogenicity Island 1 Gene mpl and Functions as a Trans-Acting sRNA in Intracellular Bacteria"

_ijms, 2024, doi:10.3390/ijms25179380_

Round 1

Reviewer 1 Report

Comments and Suggestions for Authors

Rli51 attenuates transcription of the Listeria pathogenicity is-2 land 1 gene mpl and functions as trans-acting sRNA in intracellular bacteria

by Morón-García et al.

In this interesting paper, the authors examine the regulatory mechanism and intracellular targets of a small RNA, Rli51. They find that it acts in cis to regulate transcription termination and also acts in trans as a regulatory RNA on several mRNA targets. The work adds to the growing number of examples of dual-acting RNAs and should be of wide interest to those working on RNA and pathogenesis.

In general  the experiments are well designed and executed.

I have a few small comments:

Figure 1:

1c: The statistical significance of differences in the egfp mRNA levels (bottom left graph) needs to be addressed

A double colon (::) in delta rli51::rli51 seems to indicate a plasmid-complemented strain. Usually, a double colon (::) indicates an insertion of a transposon or similar. This is confusing here.

In the legend, Δrli51 (white box) is incorrect , I think, white points is what is meant?

lines 148-9 ' the absence of rli51 seemed to have a 148 positive effect on mpl-egfp expression (Figure 1C).' This is debatable, since some of the data sets are within the experimental error bars. Again, some statistical tests of significance are needed here.

Line 192: Again, there is a similar need for statistical significance tests for the data referred to here and presented in figure 2B.

Figure 2 legend: ' (ii) Rli51 without the 17-nt-long sequence that links it to mpl'  I'm confused by the 17 nt mentioned here; should this not be 30 nt, as depicted in 2B?

Also, ' fusions driven by an inducible et promoter.' I think you mean the Tet promoter.

Figure 3 legend: ' The points represent the ratio of of each specific sRNA'.  'of' is repeated.

Line 287: It's unclear how the 6-fold enrichment number was calculated. Please clarify.

The Discussion should be shortened and the more speculative elements removed, as they tend to drag on too much and detract from an otherwise interesting paper.

Author Response

Reviewer #1:

In this interesting paper, the authors examine the regulatory mechanism and intracellular targets of a small RNA, Rli51. They find that it acts in cis to regulate transcription termination and also acts in trans as a regulatory RNA on several mRNA targets. The work adds to the growing number of examples of dual-acting RNAs and should be of wide interest to those working on RNA and pathogenesis.

In general  the experiments are well designed and executed.

  • We deeply appreciate the time and consideration taken by Reviewer #1 and his positive feedback. His comments definitively contributed to improve the quality and readability of our MS.

I have a few small comments:

  • We examined all the points raised and tried to address them the best we could. Please, see our point-by-point response below.
  1. Figure 1:
    • 1c: The statistical significance of differences in the egfp mRNA levels (bottom left graph) needs to be addressed.
      • The statistical significance of the differences could not be evaluated because the sample size was n = 2.
    • A double colon (::) in delta rli51::rli51 seems to indicate a plasmid-complemented strain. Usually, a double colon (::) indicates an insertion of a transposon or similar. This is confusing here.
      • The denomination of the strain has been replaced by Δrli51/prli51 in Figure 1.
    • In the legend, Δrli51 (white box) is incorrect, I think, white points is what is meant?
      • The Legend to Figure 1 has been amended.
  1. lines 148-9 ' the absence of rli51 seemed to have a positive effect on mpl-egfp expression (Figure 1C).' This is debatable, since some of the data sets are within the experimental error bars. Again, some statistical tests of significance are needed here.
    • That sentence was not meant to claim any strong conclusion (seemed to have…) because the sample size was n= 2. We only meant to describe a trend with no statistical support. In the revised version of the MS, the sentence has been edited in order to ascribe the observation specifically to the dataset where experimental groups do not overlap (line 150-151).
  2. Line 192: Again, there is a similar need for statistical significance tests for the data referred to here and presented in figure 2B.
    • Sample size in the 5’hly-mpl::egfp experimental group was n =2. Thus, for the analyses involving this group we could only claim the observation of a trend. In the revised version of the MS, we have de-emphasized the statement (line 194-196) and the conclusion (line 199) in this paragraph. We also restricted our observation to the “intracellular bacteria” condition (line 195), where there was not overlap in the data points between “rli51-mpl::egfp” and “5’hly-mpl::egfp” experimental groups.
  3. Figure 2 legend: ' (ii) Rli51 without the 17-nt-long sequence that links it to mpl' I'm confused by the 17 nt mentioned here; should this not be 30 nt, as depicted in 2B?
    • Corrected in the revised version of the MS.
  4. Also, ' fusions driven by an inducible et promoter.' I think you mean the Tet promoter.
    • Corrected in the revised version of the MS.
  5. Figure 3 legend: ' The points represent the ratio of of each specific sRNA'.  'of' is repeated.
    • Corrected in the revised version of the MS.
  6. Line 287: It's unclear how the 6-fold enrichment number was calculated. Please clarify.
    • To clarify this point in the main text, we have now included in the revised version of the MS that the enrichment was calculated with respect to the amount of Rli21 in the RNA pull-down performed on the control strain that expresses Rli51 without the MS2 tags. In addition, as Figure 3D has two scales on the same plot (left and right Y-axis), we now specify which one serves as a reference for each sRNA (Rli51 or Rli21) quantification. Finally, a more detailed description on how this enrichment is included in the caption of Figure 3 and in M&M section (lines 609-611).
  7. The Discussion should be shortened and the more speculative elements removed, as they tend to drag on too much and detract from an otherwise interesting paper.
    • The Discussion section has been reviewed and edited. In the revised version of the MS,
      1. we removed elements already mentioned in the Introduction (1st paragraph of the Discussion);
      2. removed the whole paragraph discussing a possible effect of Rli51 over cell surface (LPXTG) proteins and eliminated the reference to these proteins in the Abstract;
      3. removed, de-emphasized ore summarized ideas that might sound somehow speculative;
      4. edited the style, merged, or remove specific sentences, to make the text easier to read.

Reviewer 2 Report

Comments and Suggestions for Authors

This manuscript describes a careful and thorough study of transcriptional regulation that differs when extracellular and intracellular growth is examined.

Comments on the Quality of English Language

I noted these needed changes, but the whole manuscript needs to be rechecked by a native speaker. Missing commas were a major problem.

Line 59 “are”, not “is”

Line 120 Should read “Consistently, mpl RNA abundance in bacteria…”

Line 128 “Such a situation…” Also “Independent of” not “on”

Line 131 “upstream of mpl, we…” Also “ín which the rli51”

Line 132-133  “the rli51promoter, mpl-egfp expression”

Line 139 “includes” instead of “comprises” Also, “as a 5’UTR”

Line 140 “In addition to”, not “next to”

Line 142-143 ‘independently of these…factors, we”

Line 144 , line 301, Line 476-477 “To accomplish this aim,”

Line 145 “conditions, a level”

Line 154-155 I do not know what this sentence is supposed to mean.

Line156 “such regulators,”

Line 166 “These results suggest”

Line 175 “transcription, we”

Line 177 “approach, we”

Line 262 “output results, we first”

Line 329 “lay” not “lied”

Author Response

Reviewer #2:

This manuscript describes a careful and thorough study of transcriptional regulation that differs when extracellular and intracellular growth is examined.

  • We are thankful for the time and consideration taken by Reviewer #2 and his positive feedback.

Comments on the Quality of English Language

I noted these needed changes, but the whole manuscript needs to be rechecked by a native speaker. Missing commas were a major problem.

  • We appreciate the grammatical mistakes and typos identified and corrected them all in the revised version of the MS. In addition, we have carefully gone through the whole MS and put a special attention to missing commas. We trust this revision has significantly improved the quality and readability of the text.
  • Line 59 “are”, not “is”
    • Corrected in the revised version of the MS
  • Line 120 Should read “Consistently, mpl RNA abundance in bacteria…”
    • Corrected in the revised version of the MS
  • Line 128 “Such a situation…” Also “Independent of” not “on”
    • Corrected in the revised version of the MS
  • Line 131 “upstream of mpl, we…” Also “ín which the rli51”
    • Corrected in the revised version of the MS
  • Line 132-133  “the rli51promoter, mpl-egfpexpression”
    • Corrected in the revised version of the MS
  • Line 139 “includes” instead of “comprises” Also, “as a 5’UTR”
    • Corrected in the revised version of the MS
  • Line 140 “In addition to”, not “next to”
    • Corrected in the revised version of the MS
  • Line 142-143 ‘independently of these…factors, we”
    • Corrected in the revised version of the MS
  • Line 144 , line 301, Line 476-477 “To accomplish this aim,”
    • Corrected in all instances throughout the text in the revised version of the MS
  • Line 145 “conditions, a level”
    • Corrected in the revised version of the MS
  • Line 154-155 I do not know what this sentence is supposed to mean.
    • The sentence has been split and re-written (line 155-157)
  • Line156 “such regulators,”
    • Corrected in the revised version of the MS
  • Line 166 “These results suggest”
    • Corrected in the revised version of the MS
  • Line 175 “transcription, we”
    • Corrected in the revised version of the MS
  • Line 177 “approach, we”
    • Corrected in the revised version of the MS
  • Line 262 “output results, we first”
    • Corrected in the revised version of the MS
  • Line 329 “lay” not “lied”
    • Corrected in the revised version of the MS